# Regularized Gradient Boosting

**Corinna Cortes**
Google Research
New York, NY 10011
corinna@google.com

**Mehryar Mohri**
Google & Courant Institute
New York, NY 10012
mohri@google.com

**Dmitry Storcheus**
Courant Institute & Google
New York, NY 10012
dstorcheus@google.com

## Abstract

Gradient Boosting (GB) is a popular and very successful ensemble method for binary trees. While various types of regularization of the base predictors are used with this algorithm, the theory that connects such regularizations with generalization guarantees is poorly understood. We fill this gap by deriving data-dependent learning guarantees for GB used with *regularization*, expressed in terms of the Rademacher complexities of the constrained families of base predictors. We introduce a new algorithm, called RGB, that directly benefits from these generalization bounds and that, at every boosting round, applies the *Structural Risk Minimization* principle to search for a base predictor with the best empirical fit versus complexity trade-off. Inspired by *Randomized Coordinate Descent* we provide a scalable implementation of our algorithm, able to search over large families of base predictors. Finally, we provide experimental results, demonstrating that our algorithm achieves significantly better out-of-sample performance on multiple datasets than the standard GB algorithm used with its regularization.

## 1 Introduction

Ensemble methods form a powerful family of techniques in machine learning that combine multiple base predictors to create more accurate ones. These methods are often very effective in practice and can achieve a significant performance improvement over the individual base predictors [Quinlan et al., 1996, Caruana et al., 2004, Freund et al., 1996, Dietterich, 2000]. ADABOOST [Freund and Schapire, 1997] and its variants are among the most prominent ensemble methods since they are both very effective in practice and benefit from well-studied theoretical margin guarantees [Freund and Schapire, 1997, Koltchinskii and Panchenko, 2002].

Gradient Boosting (GB) [Friedman, 2001] is another popular tree-based ensemble method that has inspired a number of widely-used software libraries (e.g., XGBOOST [Chen and Guestrin, 2016], MART [Friedman, 2002], and DART [Rashmi and Gilad-Bachrach, 2015]) and has frequently ranked among the top in benchmark competitions such as *Kaggle*. But, while it is often introduced and presented differently, GB exactly coincides with AdaBoost, when the objective function used is the exponential function, as shown for example by [Schapire and Freund, 2012]. More generally, both of these algorithms are instances of *Functional Gradient Descent* [Mason et al., 2000, Grubb and Bagnell, 2011] when non-increasing convex and differentiable upper bounds on the zero-one loss are used. Viewed from the Functional Gradient Descent perspective, at every boosting step, GB seeks a predictor function $h$ that is closest to the *functional gradient* of the objective within some constrained family of base predictors $\mathcal{H}$. Specifying this base predictor family $\mathcal{H}$ such that the selected function does not overfit the gradient, as well as defining an efficient search procedure over $\mathcal{H}$ is crucial for the success of the algorithm. In most practical instances, several types of constraints are imposed to do so. As an example, for binary regression trees, XGBOOST bounds the number of leaves and the norm of the leaf values vector. This can be viewed as a *regularization*. However, to our knowledge, no theoretical analysis has been provided for these commonly-used constraints.

A natural question is whether one can derive learning guarantees that explain how this regularization on $\mathcal{H}$, and, perhaps even more general forms of constraints on functions $h \in \mathcal{H}$, are connected to the generalization performance of GB. We seek inspiration from the margin-based learning bounds given for ADABOOST [Schapire et al., 1997, Mohri et al., 2012]. These guarantees, however, do not provide a detailed analysis of the constraints on the families of tree base predictors, nor do they provide guidance on how to conduct an efficient search of these families to select a predictor during each boosting round.

We fill this gap by providing a comprehensive analysis of regularization in GB and derive learning guarantees that explain what type of regularization should be used and how. We give data-dependent learning bounds for GB with regularization, expressed in terms of the Rademacher complexities of the constrained hypotheses' sub-families, from which the base predictors are selected, as well as the ensemble mixture weights. We present a new algorithm, called RGB for Regularized Gradient Boosting, which generalizes the existing gradient boosting methods by introducing a general functional $q$-norm constraint for the families of the tree base predictors.

Our algorithm and its objective function are directly guided by the theory we develop. Our bound suggests that the *Structural Risk Minimization* principle (SRM) [Vapnik, 1992] should be used to break down $\mathcal{H}$ into subsets of varying complexities and, at each round, select a base learner $h$ from a subset that provides the best trade-off between proximity to the functional gradient and the complexity.

Applying SRM to search over subsets of $\mathcal{H}$ is challenging, since often these subsets are extremely rich, possibly infinite. An example is the families of decision trees with bounded depth used in GB. We provide a solution to the problem of expensive search and show how *Randomized Coordinate Descent* [Nesterov, 2012] can be used to search over $\mathcal{H}$ efficiently, using our generalization bounds.

Finally, this paper provides experimental results, demonstrating that our algorithm achieves significantly better out-of-sample performance than the baselines such as XGBOOST on multiple datasets. We give specific bounds, as well as the pseudocode and experimental results, for the families of binary regression trees, but our analysis can be extended to broader families of functions, such as SVMs [Cortes and Vapnik, 1995] and Deep Neural Networks [LeCun et al., 2015].

The paper is organized as follows. In Section 2, we introduce what we name a *Regularized Gradient Boosting* framework. In Section 3 we derive a Rademacher complexity bound on the families of regularized regression trees, which allows us to establish learning guarantees for Regularized Gradient Boosting. This bound directly inspire the optimization objective and the RGB algorithm presented in Section 4 that benefits from the guarantees following from the SRM principle. A non-uniform randomized search over the families of base predictors provides an efficient solution. In Section 5, we present our experimental results, which illustrate the benefits of the RGB algorithm.

## 2 Regularized Gradient Boosting

In this section, we examine the correspondence between gradient descent in functional spaces and coordinate descent in vector spaces. This connection will help us rigorously define a Regularized Gradient Boosting learning scenario and develop a scalable implementation for it.

### 2.1 Gradient Boosting as Functional Gradient Descent

Let $\mathcal{X}$ denote the input space, and let $\mathcal{F}$ be an inner product space of functions from $\mathcal{X}$ to $\mathbb{R}$. We define a restricted family of functions $\mathcal{H} \subseteq \mathcal{F}$ to be a set of base hypotheses. In a standard supervised learning scenario, the training and test points are drawn i.i.d. according to some distribution $\mathcal{D}$ over $\mathcal{X} \times \{-1, 1\}$, and $S = \{(x_1, y_1), \ldots, (x_m, y_m)\}$ is a training sample of size $m$ drawn from $\mathcal{D}^m$. In this scenario, a general boosting algorithm selects a sequence of functions $h_1, \ldots, h_T$ from $\mathcal{H}$ to minimize a certain empirical loss $\mathcal{L} \colon \mathcal{F} \mapsto \mathbb{R}$. [Friedman, 2001, Grubb and Bagnell, 2011, Mason et al., 2000, Schapire, 1999, Cortes et al., 2014]. The specification of $\mathcal{H}$ and the method of selecting each $h_t \in \mathcal{H}$ are essential for the success of the boosting algorithms. In fact, different answers to these two questions have resulted in distinct and separately-studied algorithms, such as GB, ADABOOST, and LOGITBOOST [Friedman et al., 1998].

The goal of boosting algorithms is typically to minimize an empirical loss functional:

$$\mathcal{L}(F) = \frac{1}{m} \sum_{i=1}^{m} \Phi\left(y_i, F(x_i)\right), \tag{1}$$

where $F(x) = \sum_{t=1}^{T} \alpha_t h_t(x)$ such that $\forall t \in [1, T] : h_t \in \mathcal{H}$. Popular ensemble learning algorithms, such as ADABOOST and GB, despite having originated in different research communities at different times, are particular instances of a more general algorithm, Functional Gradient Descent. The objective in Equation 1 is viewed by the Functional Gradient Descent as a functional rather than a vector-valued function, with the goal of minimizing $\mathcal{L}$ over $\mathcal{F}$ by taking steps in the direction of the steepest descent $F \leftarrow F - \eta \nabla \mathcal{L}(F)$ for some positive learning rate $\eta \in \mathbb{R}$. In the learning scenario described above, only the trace of $F$ on $x_1, \ldots, x_m$ is observable; therefore, the functional gradient of $\mathcal{L}$ is $\nabla \mathcal{L} = \left[ \frac{\partial \mathcal{L}(F)}{\partial F(x_1)}, \ldots, \frac{\partial \mathcal{L}(F)}{\partial F(x_m)} \right]$. This makes the Functional Gradient Descent update equal to $F(x_i) \leftarrow F(x_i) - \eta \frac{\partial \mathcal{L}(F)}{\partial F(x_i)}$. Of course, to make sure this functional update is well defined and to avoid over-fitting, it is natural to restrict $F(x)$ to some hypothesis set $\mathcal{H}$, which implies the following form of the functional update:

$$h = \operatorname*{argmin}_{h \in \mathcal{H}} d(\nabla \mathcal{L}, h), \tag{2}$$

where $d$ is some distance measure. This means that $h \in \mathcal{H}$ is chosen to be the closest function $h \in \mathcal{H}$ to the projection of $\nabla \mathcal{L}$ onto $\mathcal{H}$. The update in Equation 2 is a fundamental but not well-studied component of virtually all boosting methods. Simply by varying the choice of $\mathcal{H}$ and $d$, this single equation recovers most widely-used boosting algorithms.

If we restrict the optimization steps to a set of base hypotheses $\mathcal{H}$, then each step is chosen to be the function closest in the direction to the negative gradient, which means it maximizes

$$-\nabla \mathcal{L} \cdot h = -\sum_{i=1}^{m} \frac{\partial \mathcal{L}(F)}{\partial F(x_i)} h(x_i). \tag{3}$$

Particularly, if $\Phi(y_i, f(x_i)) = e^{-y_i f(x_i)}$, then the Functional Gradient Descent recovers ADABOOST, and if $\Phi(y_i, f(x_i)) = \log\left(1 + e^{-y_i f(x_i)}\right)$, then it recovers LOGITBOOST. When, instead of the negative inner product $-\nabla \mathcal{L} \cdot h$, we minimize the distance $\| - \nabla \mathcal{L}(F) - h\|_2^2$, we recover the GB algorithm.

## 2.2 Gradient Boosting as Vector Space Coordinate Descent

There is an equivalence relation between gradient descent in functional spaces and coordinate descent in vector spaces that often helps to obtain efficient algorithms for ensemble learning. At each of the $T$ steps of the Functional Gradient Descent, $\nabla \mathcal{L}$ is projected onto $\mathcal{H}$, hence the final solution $F$ can be expressed as $F_{\boldsymbol{\alpha}} = \sum_{t=1}^{T} \alpha_t h_t$ for some $\boldsymbol{\alpha} \in \mathbb{R}^T$, where $\forall 1 \leq t \leq T : h_t \in \mathcal{H}_{I_t} \subseteq \mathcal{H}$, where $\mathcal{H}_{I_t}$ indicates the subset of $\mathcal{H}$ selected at the $t$-th step. The subsets $\mathcal{H}_{I_t}$ can be viewed as coordinate blocks in $\mathcal{H}$. In this view, at boosting step $t$ a particular subspace $\mathcal{H}_{I_t}$ out of $\{\mathcal{H}_1, \ldots, \mathcal{H}_K\}$ is selected; then a base predictor $h_t \in \mathcal{H}_{I_t}$ from that subspace is added to the ensemble.

This allows switching from minimizing the loss *functional* $\mathcal{L}(F)$ to minimizing the loss function $L(\boldsymbol{\alpha}) = \mathcal{L}(F_{\boldsymbol{\alpha}})$.

$$L(\boldsymbol{\alpha}) = \frac{1}{m} \sum_{i=1}^{m} \Phi\left(y_i, \sum_{t=1}^{T} \alpha_t h_t(x_i)\right) \tag{4}$$

over the ensemble weights vector $\boldsymbol{\alpha} \in \mathbb{R}^T$. Selecting a projection $h_t$ and a step size $\alpha_t$ on the $t$-th step of the Functional Gradient Descent on $\mathcal{L}(F_{\boldsymbol{\alpha}})$ or alternatively selecting a coordinate $\alpha_t$ on the $t$-th step of the vector space coordinate descent on $L(\boldsymbol{\alpha})$ both result in the same form of the update $F_{\boldsymbol{\alpha},t} = F_{\boldsymbol{\alpha},t-1} + \alpha_t h_t$. Additionally, the full sequence of these updates for $t$ from 1 to $T$ is equal since, by the chain rule

$$\forall 1 \leq t \leq T : -\frac{\partial L(\alpha_t)}{\partial \alpha_t} = -\sum_{i=1}^{m} \frac{\partial \mathcal{L}(F_{\boldsymbol{\alpha}})}{\partial F_{\boldsymbol{\alpha}}(x_i)} h_t(x_i) = -\nabla \mathcal{L} \cdot h_t, \tag{5}$$

which means that $\min_{1 \le t \le T} -\frac{\partial L(\alpha_t)}{\partial \alpha_t}$ selected by the coordinate descent is equal to $\min_{1 \le t \le T} -\nabla \mathcal{L} \cdot h_t$ selected by Functional Gradient Descent.

This equivalence illustrates two important points. First, coordinate descent methods can be used to provide efficient numerical solutions for boosting. Second, the proper construction of the subsets $\mathcal{H}_t$ such that $h_t \in \mathcal{H}_{I_t} \subseteq \mathcal{H}$ is crucial for the success of boosting algorithms. We rely on this equivalence when presenting a coordinate-descent-style algorithm for minimizing the regularized boosting objective that scales well to large families of base predictors.

## 2.3   Regularized Gradient Boosting

In this subsection, we describe the main novelty of our work – the analysis of regularization applied to GB. We formulate what we name a *Regularized Gradient Boosting* framework and show the subtle connection between the regularization and the properties of $\mathcal{H}_k \subseteq \mathcal{H}$. As we shall see, the regularization terms are not explicitly introduced in the definition of the objective, but only in the definition of an approximation to the functional gradient.

While the *unregularized* projection step, as in Equation 2, has been extensively studied for GB, the fundamental theory of the regularization commonly used is missing. However, a number of empirical studies and software frameworks [Sun et al., 2014, Chen and Guestrin, 2016] indicate that introducing regularization to this step is extremely beneficial. For example, the popular XGBOOST library, dedicated to boosted decision trees, regularizes the norm of the leaf values, as well as the number of leafs. We are filling this gap by providing a theory that links regularization with learning guarantees for GB algorithms.

For a convex function $\Omega \colon \mathcal{F} \mapsto \mathbb{R}$, a closed subspace $\mathcal{H} \subseteq \mathcal{F}$ and $\beta \in \mathbb{R}_+$, let the Regularized Gradient Boosting step be defined by

$$h = \operatorname*{argmin}_{h \in \mathcal{H}} d(\nabla \mathcal{L}, h) + \beta \Omega(h). \tag{6}$$

Given the convexity of $\Omega$, this step is equivalent to $h = \operatorname{argmin}_{h \in \widehat{\mathcal{H}}} d(\nabla \mathcal{L}, h)$, where $\widehat{\mathcal{H}} = \mathcal{H} \cap \{h \colon \Omega(h) \le \beta\}$. Such a reduction illustrates the subtle, yet extremely important, connection between regularization and the definition of hypothesis set $\mathcal{H}$. The equivalence between vector space coordinate descent and Functional Gradient Descent presented in Section 2, meaning that both of these methods iteratively select the same sequence of functions $h_t \in \mathcal{H}_{I_t} \subseteq \mathcal{H}$, suggests that a natural way to use regularization for boosting is to define $\mathcal{F} = \operatorname{conv}(\cup_{k=1}^K \mathcal{H}_k)$, where $\mathcal{H}_k = \{h \colon \theta_{k-1} < \Omega(h) \le \theta_k\}$ are disjointed sets of functions for a set of parameters $[\theta_1, \ldots, \theta_K]$. Note that, with this formulation, the regularization is not in the objective function; instead the search for the gradient approximation is constrained by a regularization.

We show, in the following section, that such a definition of $\mathcal{F}$ allows us to obtain margin-based learning guarantees for the Regularized Gradient Boosting that are dependent on the complexities of each individual $\mathcal{H}_k$.

## 3   Learning Guarantees

As described in the previous section, by projecting the functional gradient onto $\mathcal{F} = \operatorname{conv}(\cup_{k=1}^K \mathcal{H}_k)$ at each step, we are able to learn an ensemble function $f = \sum_{t=1}^T \alpha_t f_t \in \mathcal{F}$, where the $\mathcal{H}_k$s are families of functions with varying complexity. Thus, it is natural to seek learning guarantees depending on the properties of each $\mathcal{H}_k$ and the mixture weight vector $\boldsymbol{\alpha} = [\alpha_1, \ldots, \alpha_T]$.

The first margin bound based on the VC-dimension for ensembles $\sum_{t=1}^T \alpha_t f_t$ was given by Freund and Schapire [1997]. Later, tighter data-dependent bounds in terms of the Rademacher complexity of the underlying function class $\mathcal{H}$ were given by Koltchinskii and Panchenko [2002], see also [Mohri et al., 2018]. For the specific case where $\mathcal{H} = \operatorname{conv}(\cup_{k=1}^K \mathcal{H}_k)$, Rademacher complexity-based guarantees were given in [Cortes et al., 2014]. In this section, we will use these theoretical results to derive margin-based guarantees based on the Rademacher complexities of the families of regularized decision trees $\mathcal{H}_k$ and the mixture weights $\boldsymbol{\alpha}$. The bounds that we show, being data-dependent, will not only fill the missing generalization theory for the existing gradient tree boosting frameworks but also motivate a new scalable learning algorithm for the Regularized Gradient Boosting framework, called RGB, in Section 4.

Here, we restrict our analysis to the hypothesis families $\mathcal{H}_k$ of regression trees. However, our results can be extended to other families, such as kernel-based hypotheses and neural networks, so long as the sample Rademacher complexities of these families can be bounded.

Each leaf $l$ in a regression tree contains a real-valued number $w_l$ providing the output value of the tree for any sample point allocated to that leaf; thus, we let $\mathbf{w}$ be a vector of stacked leaf values. The function computed by a regression tree can thus be represented by $h(x) = \sum_{l \in \text{leaves(h)}} w_l \mathbb{I}\{x \in \text{leaf}_l\}$, where $\mathbb{I}\{x \in \text{leaf}_l\}$ is the indicator function for sample point $x \in \mathbb{R}^d$ being allocated to $\text{leaf}_l$; this value $h(x)$ can be used for classification in a straightforward manner by thresholding.

The node partition functions in binary regression trees are of the form $[x]_j \leq \theta$ for some feature index $j \in [1, d]$ and $\theta \in \mathbb{R}$, which means that if $[x_i]_j \leq \theta$ for a sample point $x_i \in \mathbb{R}^d$, then $x_i$ is allocated to the left subtree and to the right subtree otherwise. Let $\mathcal{H}_{n,\lambda,q}$ be the set of all *regularized* binary regression trees with the number of internal nodes bounded by $n$ and a leaf values vector $\mathbf{w}$ such that $\|\mathbf{w}\|_q \leq \lambda$, $q \geq 1$. Special instances of these families of trees are widely used in practice. For example, $\mathcal{H}_{n,\lambda,1}$ and $\mathcal{H}_{n,\lambda,2}$ are implemented in XGBOOST and frequently used in practice.

**Theorem 1.** *For any sample $S = (x_1, \ldots, x_m)$, the empirical Rademacher complexity of a hypothesis set $\mathcal{H}$ is defined by $\widehat{\mathfrak{R}}_S(\mathcal{H}) = \mathbb{E}_{\boldsymbol{\sigma}} \left[ \sup_{h \in \mathcal{H}} \sum_{i=1}^m \sigma_i h(x_i) \right]$, where, $\sigma_i s$, $i \in [m]$, are independent uniformly distributed random variables taking values in $\{-1, 1\}$. Let $d$ be the input data dimension. The following upper bound holds for the empirical Rademacher complexity of $\mathcal{H}_{n,\lambda,q}$:*

$$\widehat{\mathfrak{R}}^S(\mathcal{H}_{n,\lambda,q}) \leq \lambda \sqrt{\frac{(4n+2)\log_2(d+2)\log(m+1)}{m}}.$$

The proof of Theorem 1 is given in the Appendix. This bound shows how the empirical Rademacher complexity of the regularized decision trees depends both on on the number of internal nodes $n$ and the upper bound $\lambda$ on the $q$-norm of leaf values.

Using this bound, we can now derive our margin-based learning guarantees for the family $\mathcal{F}$. Let $R(f)$ denote the binary classification error of $f \in \mathcal{F}$, $R(f) = \mathbb{E}_{(x,y)\sim\mathcal{D}} \mathbb{I}\{yf(x) \leq 0\}$, and $R_\rho(f)$ its empirical $\rho$-margin loss for a sample $S$, $R_\rho(f) = \mathbb{E}_{(x,y)\sim\mathcal{D}} \mathbb{I}\{yf(x) \leq \rho\}$. Let $\widehat{R}_\rho(f) = \mathbb{E}_{(x,y)\sim S} \mathbb{I}\{yf(x) \leq \rho\}$.

**Theorem 2.** *Fix $\rho > 0$. Let $\mathcal{H}_k = \mathcal{H}_{n_k, \lambda_k, q_k}$, where $(n_k)$, $(\lambda_k)$ are sequences of constraints on the number of internal nodes $n$ and the leaf vector norm $\|\mathbf{w}\|_q$. Define $\mathcal{F} = \text{conv}(\cup_{k=1}^K H_k)$. Then, for any $\delta > 0$, with probability at least $1 - \delta$ over the draw of a sample $S$ of size $m$, the following inequality holds for all $f = \sum_{t=1}^T \alpha_t h_t \in \mathcal{F}$:*

$$R(f) \leq \widehat{R}_{S,\rho}(f) + \frac{4}{\rho} \sum_{t=1}^T \alpha_t \lambda_{I_t} \sqrt{\frac{(4n_{I_t}+2)\log_2(d+2)\log(m+1)}{m}} + C(m, K),$$

*where $I_t$ is the index of the subclass selected at time $t$ and $C(m, K) = O\left( \sqrt{\frac{\log(K)}{\rho^2 m}} \log \left[ \frac{\rho^2 m}{\log(K)} \right] \right)$.*

The proof of Theorem 2 is given in the Appendix. The generalization bound of Theorem 2 motivates a specific algorithm for Regularized Gradient Boosting, described and discussed in the next section.

## 4  Algorithm

The multiplicative structure of the bound in Theorem 2 with respect to the mixture weights $[\alpha_1, \ldots, \alpha_T]$ and the complexities $\mathcal{H}_{I_t}$ suggests the use of these complexities (or their upper bounds) in the regularization $\Omega(h)$. Additionally, one may upper-bound the empirical loss function of $u \mapsto \mathbb{I}\{u \leq 0\}$ in Theorem 2, leading to the following objective:

$$L(\boldsymbol{\alpha}) = \frac{1}{m} \sum_{i=1}^m \Phi\left( y_i, \sum_{t=1}^T \alpha_t h_t(x_i) \right) + \beta \sum_{t=1}^T |\alpha_t| \lambda_{I_t} \sqrt{\frac{(4n_{I_t}+2)\log_2(d+2)\log(m+1)}{m}}. \quad (7)$$

Minimizing the function with vector space coordinate descent is equivalent to solving for a projection at each Functional Gradient Descent step of the form

$$h_t = \underset{h \in \mathcal{H}}{\text{argmin}}\, d(\nabla\mathcal{L}, h) + \beta \sum_{k=1}^K \lambda_k \sqrt{\frac{(4n_k+2)\log_2(d+2)\log(m+1)}{m}} \mathbb{I}\{h \in \mathcal{H}_k\}. \quad (8)$$

In this section we will devise an algorithm for minimizing the regularized objective $L(\boldsymbol{\alpha})$, called RGB, that is able to scale to large families of base predictors.

## 4.1 Randomized Coordinate Descent

The practical challenge of building an ensemble of base predictors in the Regularized Gradient Boosting scenario is to both define the hypothesis sets $\mathcal{H}_k$ and implement an efficient search across these sets to select the best update direction $h_t$, at each optimization step. Applying coordinate descent to the objective in Equation 7 may be feasible for finite hypothesis sets; however, we are often required to work with infinite spaces of subfamilies of functions. A typical example would be one where each subfamily is a decision tree with a fixed topology and fixed leaf values. It is common to resort to heuristics or to discretize the search space to define an approximate search.

To solve the problem of an extensive search over $\mathcal{H}_k$, we propose a novel method for boosting updates using randomization applied to the functional space. Random selection of base learners for GB in the context of *Randomized Coordinate Descent* has been shown to be successful in practice. For example, [Lu and Mazumder, 2018] demonstrated that uniform sampling helps make the search over base hypothesis classes more scalable, gave favorable convergence guarantees for this method. Nesterov [2012] introduced probabilistic convergence guarantees for Randomized Coordinate Descent expressed in terms of the local smoothness properties of the objective and suggested a distribution to sample the coordinates.

Inspired by the analysis of Nesterov [2012], our work is the first one to provide a fundamentally-justified method of searching over the subspaces $\mathcal{H}_k$, an algorithm that is both scalable and admits convergence guarantees. The RGB algorithm picks at each round at random a subset $\{\mathcal{H}_{t_1}, \ldots, \mathcal{H}_{t_S}\}$. Given a meaningful distribution over $\mathcal{H}$ that captures the steepness of the objective $L(\boldsymbol{\alpha})$ within each of these subsets, RGB is able to learn an ensemble of functions from families $\mathcal{H}_k$ of varying complexity. In the following, we show how to apply the Randomized Coordinate Descent method, as in [Nesterov, 2012], to the objective 7.

## 4.2 Lipschitz-Continuous Gradients

Consider the problem of minimizing $L(\boldsymbol{\alpha})$ as in Equation 7. The following lemma describes the continuity properties of the partial derivatives of $L(\boldsymbol{\alpha})$, which are needed for the application of Randomized Coordinate Descent.

**Lemma 3.** *Assume that $\Phi(y, h)$ is differentiable with respect to the second argument, and that $\frac{\partial \Phi}{\partial h}$ is $C_\Phi(y)$-Lipschitz with respect to the second argument, for any fixed value $y$ of the first argument. For all $k \in [0, K]$, define $L'_k(\boldsymbol{\alpha}) = \frac{\partial L}{\partial \alpha_k}$. Then, $L'_k(\boldsymbol{\alpha})$ is $C_k$-Lipschitz with $C_k$ bounded as follows:*

$$C_k \leq \frac{1}{m} \sum_{i=1}^{m} h_k^2(x_i) C_\Phi(y_i). \tag{9}$$

Randomized Coordinate Descent samples the $k$-th coordinate with probability $p_k = C_k / \sum_{k=1}^{K} C_k$. The convergence guarantees for this procedure are given in [Nesterov, 2012] as a function of the Lipschitz constants $C_k$.

We can further give upper bounds for the Lipschitz constants above to avoid the computation of the sums $\sum_{i=1}^{m} h_k^2(x_i)$. If we introduce the vectors $\mathbf{h}_k$ and $\mathbf{C}_\Phi$ in $\mathbb{R}^m$ such that $[\mathbf{h}_k]_i = h_k^2(x_i)$ and $[\mathbf{C}_\Phi]_i = C_\Phi(y_i)$, then, by Hölder's inequality,

$$C_k \leq \frac{1}{m} \sum_{i=1}^{m} h_k^2(x_i) C_\Phi(y_i) = \frac{1}{m} \mathbf{h} \cdot \mathbf{C}_\Phi \leq \frac{1}{m} \|\mathbf{h}\|_r \|\mathbf{C}_\Phi\|_q, \tag{10}$$

where $\frac{1}{r} + \frac{1}{q} = 1$. Various $(r, q)$-dual norms can be used, depending on the computational constraints and the complexity of the hypothesis classes for the application of the Randomized Coordinate Descent method. For example, using $\|\mathbf{h}\|_1$ and $\|\mathbf{C}_\Phi\|_\infty$ gives the following upper bound: $C_k \leq \frac{1}{m} \left[ \max_{1 \leq i \leq m} C_\Phi(y_i) \right] \sum_{i=1}^{m} h_k^2(x_i)$.

The developed generalization bounds imply the Lipschitz constants and thus define the Randomized Coordinate Descent steps for the minimization of $L(\boldsymbol{\alpha})$, controlling its convergence. To illustrate this

---

**Algorithm 1** RGB. Input: $\boldsymbol{\alpha} = 0, F = 0$

---

1: **for** $t \in [1, T]$ **do**
2: $\quad [t_1, \cdots, t_S] \leftarrow P$
3: $\quad$ **for** $s \in [1, S]$ **do**
4: $\quad\quad h_s \leftarrow \operatorname{argmin}_{h \in \mathcal{H}_{t_s}} \frac{1}{m} \sum_{i=1}^{m} \Phi\big(y_i, F - \frac{1}{C_{t_s}} L'_{t_s}(\boldsymbol{\alpha})h\big)$
5: $\quad$ **end for**
6: $\quad s^\star = \operatorname{argmin}_{s \in [1,S]} \big[\frac{1}{m} \sum_{i=1}^{m} \Phi\big(y_i, F - \frac{1}{C_{t_s}} L'_{t_s}(\boldsymbol{\alpha})h_s\big) + \beta\Omega(h_{t_s})\big]$
7: $\quad \boldsymbol{\alpha} \leftarrow \boldsymbol{\alpha} - \frac{1}{C_{s^\star}} L'_{s^\star}(\boldsymbol{\alpha})\mathbf{e}_{t_{s^\star}}$
8: $\quad F \leftarrow F - \frac{1}{C_{s^\star}} L'_{s^\star}(\boldsymbol{\alpha})h_{s^\star}$
9: **end for**

---

point, the convergence rate stated in [Nesterov, 2012] is as follows:

$$\mathbb{E}_{t-1}\big[L(\boldsymbol{\alpha}_t)\big] - L(\boldsymbol{\alpha}_\star) \leq \frac{2}{t+1}\bigg(\sum_{j=1}^{K} C_j\bigg) R_0^2(\boldsymbol{\alpha}_0), \tag{11}$$

where $\boldsymbol{\alpha}_0$ is the starting point, $\boldsymbol{\alpha}_\star$ is the global minimizer of $L(\boldsymbol{\alpha})$ and $R_0(\boldsymbol{\alpha}_0)$ is the size of the initial level set of the objective. The conditional expectation is taken over the random choice of the next coordinate. The regularization applied to the base predictor families in our Regularized Gradient Boosting Framework implies the bounds on $C_k$, thus controlling the convergence of the algorithm.

### 4.3 Pseudocode

The pseudocode of our RGB algorithm is given in Algorithm 1. The algorithm seeks to minimize the objective given in Equation 7, using Randomized Coordinate Descent. Let $P$ be a discrete probability distribution over $[1, K]$ with $p_k = C_k / \sum_{j=1}^{K} C_j$. Equivalently, $P$ is the distribution over the base hypothesis sets $\mathcal{H}_1, \cdots, \mathcal{H}_K$. At each draw from $P$, we select a sample $\mathcal{H}_{t_1}, \cdots, \mathcal{H}_{t_S}$ of size $S$ and, from this sample, select one function that provides the best trade-off in the decrease in objective $L(\boldsymbol{\alpha})$ and the complexity bound of Theorem 1 of the underlying hypotheses family.

The local optimization procedure in Line 6 is an extra step required in the coordinate descent procedure to select a single function from $\mathcal{H}_{t_s}$. The step in Line 8 is required to select, out of $S$ sampled directions, the one with the best trade-off between sample fit and complexity bounds. Note that the evaluation of sampled candidates in Line 5 can be done in parallel, making the time of RGB per thread comparable to that of standard GB. More specifically, given a fixed sample of $S$ coordinates, the runtime of one RGB round is equal to the runtime of $S$ rounds of GB when the same subroutine is used for tree splitting.

## 5 Experiments

In this section, we present the results of experiments with our RGB algorithm. We restrict our attention to learning an ensemble of the regularized regression trees as defined in the family $\mathcal{H}_{n,\lambda,q}$, and to simplify the presentation, we let $q = 2$, although similar experiments can be easily done for other norms. For the complexity of these base classifiers we use the bound derived in Theorem 1.

To define the subfamilies of base learners we impose a grid of size 7 on the maximum number of internal nodes $n \in \{2, 4, 8, 16, 32, 64, 256\}$ and a grid of size 7 on $\lambda \in \{0.001, 0.01, 0.1, 0.5, 1, 2, 4\}$. For each element from the Cartesian product of these grids, we assign $(n_k, \lambda_k)$, thus defining the base families $\mathcal{H}_{n_k,\lambda_k,2}$ and $\mathcal{F} = \operatorname{conv}\big(\cup_{k=1}^{49} \mathcal{H}_{n_k,\lambda_k,2}\big)$. Given such a decomposition of the functional space, we directly minimize the regularized objective in Equation 7 using Randomized Coordinate Descent with the distribution over the coordinate blocks as described above. We use the logistic loss as the per-instance loss $\Phi$. For a given training sample, we normalize the regularization $\Omega(h)$ to be in $[0, 1]$ and tune the RGB parameter $\beta$ using a grid search over $\beta \in \{0.001, 0.01, 0.1, 0.3, 1\}$.

Section 4 describes multiple ways to bound the coordinate-wise Lipschitz constants of the derivative of the objective function, resulting in various coordinate sampling distributions for the Randomized Coordinate Descent. For our experiments, and specifically to the families $\mathcal{H}_{n_k,\lambda_k,2}$ bound the

**Table 1:** Experimental Results

| Error % | sonar | cancer | diabetes | ocr17 | ocr49 | mnist17 | mnist49 | higgs |
|---|---|---|---|---|---|---|---|---|
| | | | | RGB | | | | |
| Mean | 26.94 | 5.19 | 28.86 | 0.90 | 3.10 | 0.43 | 1.53 | 28.60 |
| (Std) | (2.10) | (0.97) | (4.85) | (0.45) | (0.69) | (0.10) | (0.38) | (0.41) |
| | | | | GB | | | | |
| Mean | 28.64 | 6.14 | 28.39 | 1.35 | 3.50 | 0.55 | 1.66 | 29.11 |
| (Std) | (2.13) | (0.94) | (5.08) | (0.52) | (0.65) | (0.11) | (0.32) | (0.37) |
| | | | One-tailed, paired sample t-test | | | | | |
| Signif. | 5% | 5% | - | 2.5% | 2.5% | 2% | 5% | 2.5% |

Lipschitz constants by $C_k \leq \lambda_k \left[ \max_{1 \leq i \leq m} C_\Phi(y_i) \right]$, which implies that the $k$-th coordinate is sampled with probability $p_k = \lambda_k / \sum_{j=1}^{K} \lambda_j$, since the $\max_{1 \leq i \leq m} C_\Phi(y_i)$ terms cancel out (see Lemma 4 in the Appendix for the derivation of this bound).

As a comparison, we run the standard GB, using the XGBOOST library with $\ell_2$ regularization on the vector of leaf scores $\mathbf{w}$. We use grid search to tune the hyperparameters of XGBOOST with a grid that makes the families of trees explored comparable to the $\mathcal{H}$ defined for RGB above. Specifically, we let the $\ell_2$ norm regularization parameter be in $\{0.001, 0.01, 0.1, 0.5, 1, 2, 4\}$, the maximum tree depth parameter in $\{1, 2, 3, 4, 5, 6, 7\}$, and the learning rate parameter in $\{0.001, 0.01, 0.1, 0.5, 1\}$. Both GB and RGB are run for $T = 100$ boosting rounds. The hyperparameters are chosen via 5-fold cross-validation, and the standard errors for the best set of hyperparameters reported.

Table 1 shows the classification errors on the test sets for the UCI datasets studied, for both RGB and GB, see Table 2 in the appendix for details on the dataset. A one-tailed, paired sample t-test on the pairs of results from the different trials demonstrate that these results are in general significant at a 5% level or better. Only for one of the dataset, `diabetes` with an input dimension of just 8, we do not observe an improvement of RGB over GB. One natural hypothesis is that the larger the input dimension, the more the need for proper regularization of the binary regression trees forming the base learner, and the larger the advantage of the RGB algorithm.

In general, the results demonstrate that by randomly taking steps into coordinates that correspond to subspaces $\mathcal{H}_t$ with a theoretically justified distribution, RGB can explore larger hypothesis families more efficiently that the baseline methods. Furthermore, compared to baselines that operate on the same hypotheses space $\mathcal{H}$, by optimizing for the trade-off between sample fit and functional subclass complexity, RGB can reduce over-fitting, thereby achieving greater test accuracy on multiple datasets.

## 6 Conclusion

We introduced and analyzed a general framework of Regularized Gradient Boosting, for which we also devised an effective algorithm, RGB. In this framework, regularization is not directly incorporated as a term in the loss function. Instead, its definition affects each boosting step by restricting the search for the gradient approximation to a constrained subset of base functions. Our analysis is based upon strong margin-based Rademacher complexity learning guarantees. These bounds suggest a natural approach for our optimization solution, which consists of dividing the space of base learners into subfamilies of increasing complexity. For the special case of binary regression trees, we derived explicit Rademacher complexity bounds that we subsequently exploit in the definition of our RGB algorithm. Randomization over the subfamilies of base functions allows us to scale our algorithm to large families of base predictors. Our experimental results suggest improved performance, thanks to a more efficient and theoretically motivated exploration of large function spaces without over-fitting. Also, as already stated, the run-times of the algorithms are comparable, thereby making RGB a strong alternative to XGBOOST. Finally, our analysis can be extended in a similar way to that of boosting with other families of base predictors, such as kernel-based hypothesis sets and Deep Neural Networks.

## Acknowledgments

We thank our colleagues Natalia Ponomareva and Vitaly Kuznetsov for insightful discussions and feedback. This work was partly supported by NSF CCF-1535987, NSF IIS-1618662, and a Google Research Award.

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
