[Supplementary Material]

# A  Appendix

 ## A.1  Proof of Theorem 1

**Theorem 1.** For any sample $S = (x_1, \ldots, x_m)$, the empirical Rademacher complexity of a hypothesis set $\mathcal{H}$ is defined by $\widehat{\mathfrak{R}}_S(\mathcal{H}) = \mathbb{E}_{\boldsymbol{\sigma}} \left[ \sup_{h \in \mathcal{H}} \sum_{i=1}^m \sigma_i h(x_i) \right]$, where, $\sigma_i$s, $i \in [m]$, are independent uniformly distributed random variables taking values in $\{-1, 1\}$. The following upper bound holds for the empirical Rademacher complexity of $\mathcal{H}_{n,\lambda,q}$:

$$\widehat{\mathfrak{R}}^S(\mathcal{H}_{n,\lambda,q}) \leq \lambda \sqrt{\frac{(4n+2)\log_2(d+2)\log(m+1)}{m}},$$

where $d$ is input data dimension.

*Proof.* For the purpose of this proof, let $\mathcal{H}_n$ be the family of binary decision trees with leaf values $w_j \in \{-1, +1\}$. We use the regularization in the family $\mathcal{H}_{n,\lambda,q}$ and the connection to the family $\mathcal{H}_n$ in the proof below. Additionally, let $r \geq 1$ such that $\frac{1}{r} + \frac{1}{q} = 1$, meaning that the $r-$norm is the dual to the $q-$norm. To aid the presentation in the proof, we are going to define a vector $\widehat{\boldsymbol{\sigma}}$ s.t. $[\widehat{\boldsymbol{\sigma}}]_j = \sum_{x_i \in \text{leaf}_j} \sigma_i$, the $j$-th coordinate of which contains the sum of the Rademacher variables that correspond to the sample points that fall within $j$-th leaf of a tree $h$.

$$\widehat{\mathfrak{R}}^S(\mathcal{H}_{n,\lambda,q}) = \frac{1}{m} \mathbb{E}_{\sigma} \left[ \sup_{h \in \mathcal{H}_{n,\lambda,q}} \left[ \sum_{n=1}^m \sigma_n h(x_n) \right] \right] \tag{12}$$

$$= \frac{1}{m} \mathbb{E}_{\sigma} \left[ \sup_{h \in \mathcal{H}_{n,\lambda,q}} \left[ \widehat{\boldsymbol{\sigma}} \cdot \mathbf{w} \right] \right] \tag{13}$$

$$\leq \frac{1}{m} \mathbb{E}_{\sigma} \left[ \sup_{h \in \mathcal{H}_{n,\lambda,q}} \|\widehat{\boldsymbol{\sigma}}\|_r \|\mathbf{w}\|_q \right] \tag{14}$$

$$\leq \frac{\lambda}{m} \mathbb{E}_{\sigma} \left[ \sup_{h \in \mathcal{H}_n} \|\widehat{\boldsymbol{\sigma}}\|_r \right] \tag{15}$$

$$\leq \frac{\lambda}{m} \mathbb{E}_{\sigma} \left[ \sup_{h \in \mathcal{H}_n} \|\widehat{\boldsymbol{\sigma}}\|_1 \right] \tag{16}$$

$$= \frac{\lambda}{m} \mathbb{E}_{\sigma} \left[ \sup_{h \in \mathcal{H}_n} \sum_{i=1}^n |[\widehat{\boldsymbol{\sigma}}]_i| \right] \tag{17}$$

$$= \frac{\lambda}{m} \mathbb{E}_{\sigma} \left[ \sup_{h \in \mathcal{H}_n} \sum_{l \in \text{leaves(h)}} \left| \sum_{i=1}^m \sigma_i 1_{\{x_i \in l\}} \right| \right] \tag{18}$$

$$\leq \frac{\lambda}{m} \mathbb{E}_{\sigma} \left[ \sup_{h \in \mathcal{H}_n, s_l \in \{+1,-1\}} \sum_{l \in \text{leaves(h)}} s_l \sum_{i=1}^m \sigma_i 1_{\{x_i \in l\}} \right] \tag{19}$$

$$= \frac{\lambda}{m} \mathbb{E}_{\sigma} \left[ \sup_{h \in \mathcal{H}_n, s_l \in \{+1,-1\}} \sum_{i=1}^m \sigma_i \sum_{l \in \text{leaves(h)}} s_l 1_{\{x_i \in l\}} \right] \tag{20}$$

$$\leq \lambda \sqrt{\frac{(4n+2)\log_2(d+2)\log(m+1)}{m}} \tag{21}$$

Where $n$ is the number of internal nodes, and $d$ is the input data dimension. The inequality (14) is a direct application of the Hölder's inequality for dual norms. The inequality (16) uses $\|\cdot\|_r \leq \|\cdot\|_1$. The equality (18) directly follows from the definition of $\widehat{\boldsymbol{\sigma}}$. The last inequality (21) follows from the fact that the VC-dimension of binary classification trees can be bounded by $(2n+1)\log_2(d+2)$ Mohri et al. [2012] and a direct application of Massart's lemma Massart and Picard [2007]. □

## A.2 Proof of Theorem 2

**Theorem 2.** Fix $\rho > 0$. Let $\mathcal{H}_k = \mathcal{H}_{n_k, \lambda_k, q_k}$, where $(n_k)$, $(\lambda_k)$ are sequences of constraints on the number of internal nodes $n$ and the leaf vector norm $\|\mathbf{w}\|_q$. Define $\mathcal{F} = \mathrm{conv}(\cup_{k=1}^K H_k)$. Then, for any $\delta > 0$, with probability at least $1 - \delta$ over the draw of a sample $S$ of size $m$, the following inequality holds for all $f = \sum_{t=1}^T \alpha_t h_t \in \mathcal{F}$:

$$R(f) \leq \widehat{R}_{S,\rho}(f) + \frac{4}{\rho} \sum_{t=1}^T \alpha_t \lambda_{I_t} \sqrt{\frac{(4n_{I_t} + 2)\log_2(d+2)\log(m+1)}{m}} + C(m, K),$$

where $I_t$ is the index of the subclass selected at time $t$ and $C(m, K) = O\left( \sqrt{\frac{\log(K)}{\rho^2 m} \log\left[ \frac{\rho^2 m}{\log(K)} \right]} \right)$.

*Proof.* For this proof we are going to make use of the generalization bounds for broad families of real-valued functions given in Theorem 1 of [Cortes et al., 2014]. Adapted to our notation, it states that for any $f$ from a family of real-valued functions $\mathcal{F}$ that is equal to the convex hull of $\cup_{k=1}^K \mathcal{H}_k$, for any $\delta > 0$ with probability at least $1 - \delta$ over the choice of sample $S \sim \mathcal{D}^m$, the following generalization bound holds:

$$R(f) \leq \widehat{R}_{S,\rho}(f) + \frac{4}{\rho} \sum_{t=1}^T \alpha_t \mathfrak{R}_m(\mathcal{H}_t) + \frac{2}{\rho} \sqrt{\frac{\log K}{m}} + \sqrt{\left\lceil \frac{4}{\rho^2} \log\left( \frac{\rho m^2}{\log K} \right) \right\rceil \frac{\log K}{m} + \frac{\log(\frac{2}{\delta})}{2m}}.$$

where $\alpha_t$ is are the weights that represent $f$ in the convex hull of $\cup_{k=1}^K \mathcal{H}_k$, that is $f = \sum_{t=1}^T \alpha_t h_t$ s.t. $\boldsymbol{\alpha} = [\alpha_1, \ldots, \alpha_T]$ is in the simplex $\Delta$. This bound is directly applicable to the Regularized Gradient Boosting that we define, since at each boosting round, the algorithm selects a base predictor $h_t \in \mathcal{H}_t$, and multiplies it by a coefficient $\alpha_t$. Thus, after $T$ boosting rounds, we will have obtained an ensemble $f$ such that $f = \sum_{t=1}^T \alpha_t h_t \in \mathrm{conv}(\cup_{k=1}^K \mathcal{H}_k)$ and $\boldsymbol{\alpha}$ directly in the simplex $\Delta$.

Applying the Rademacher complexity bound on the regularized families of regression trees $\mathcal{H}_{n,\lambda,q}$ that we derived in Theorem 1 and noting that

$$\frac{2}{\rho} \sqrt{\frac{\log K}{m}} + \sqrt{\left\lceil \frac{4}{\rho^2} \log\left( \frac{\rho m^2}{\log K} \right) \right\rceil \frac{\log K}{m} + \frac{\log(\frac{2}{\delta})}{2m}} = O\left( \sqrt{\frac{\log(K)}{\rho^2 m} \log\left[ \frac{\rho^2 m}{\log(K)} \right]} \right) \quad (22)$$

We obtain the expression for the bound in Theorem 2. $\qquad\square$

## A.3 Proof of Lemma 3

**Lemma 3.** Assume that $\Phi(y, h)$ is differentiable with respect to the second argument, and that $\frac{\partial \Phi}{\partial h}$ $C_\Phi(y)$-Lipschitz with respect to the second argument, for any fixed value $y$ of the first argument. for all $k \in [0, K]$, define $L_k'(\boldsymbol{\alpha}) = \frac{\partial L}{\partial \alpha_k}$. Then, $L_k'(\boldsymbol{\alpha})$ is Lipschitz-continuous with the corresponding Lipschitz constants $C_k$ bounded as follows:

$$C_k \leq \frac{1}{m} \sum_{i=1}^m h_k^2(x_i) C_\Phi(y_i). \quad (23)$$

*Proof.* The $k$-th derivative of $L(\boldsymbol{\alpha})$ is equal to (except $\alpha_k = 0$):

$$L_k'(\boldsymbol{\alpha}) = \frac{1}{m} \sum_{i=1}^m \frac{\partial \Phi}{\partial h}\left( y_i, \sum_{t=1}^T \alpha_t h_t(x_i) \right) h_k(x_i) + c_k, \quad (24)$$

where $c_k = \beta \lambda_k \sqrt{\frac{(4n_k+2)\log_2(d+2)\log(m+1)}{m}}$. Let $\mathbf{e}_k$ be the $k$-th standard basis vector, then

$$\left| L'_k(\boldsymbol{\alpha}) - L'_k(\boldsymbol{\alpha} + \delta \mathbf{e}_k) \right| = \left| \frac{1}{m} \sum_{i=1}^{m} h_k(x_i) \left[ \frac{\partial \Phi}{\partial h}\left(y_i, \sum_{t=1}^{T} \alpha_t h_t(x_i)\right) - \frac{\partial \Phi}{\partial h}\left(y_i, \sum_{t=1}^{T} \alpha_t h_t(x_i) + \delta h_k(x_i)\right) \right] \right|$$

$$\leq \frac{1}{m} \sum_{i=1}^{m} |h_k(x_i)| \left| \frac{\partial \Phi}{\partial h}\left(y_i, \sum_{t=1}^{T} \alpha_t h_t(x_i)\right) - \frac{\partial \Phi}{\partial h}\left(y_i, \sum_{t=1}^{T} \alpha_t h_t(x_i) + \delta h_k(x_i)\right) \right|$$

$$= \frac{1}{m} \sum_{i=1}^{m} |h_k(x_i)| \left| \frac{\partial \Phi}{\partial h}\left(y_i, f\right) - \frac{\partial \Phi}{\partial h}\left(y_i, f + \delta h_k(x_i)\right) \right|$$

$$\leq \frac{1}{m} \sum_{i=1}^{m} |h_k(x_i)| C_\Phi(y_i) |h_k(x_i)| |\delta|$$

$$= \frac{1}{m} \sum_{i=1}^{m} h_k^2(x_i) C_\Phi(y_i) |\delta|$$

Thus, $L'_k(\boldsymbol{\alpha})$ is Lipschitz-continuous with the corresponding Lipschitz constant bounded by $\frac{1}{m} \sum_{i=1}^{m} h_k^2(x_i) C_\Phi(y_i)$.

$\square$

### A.4   Proof of Lemma 4

**Lemma 4.** . *For each $k \in [0, K]$ let $\mathcal{H}_{n_k, \lambda_k, 2}$ be the family of regularized regression trees with $\|\mathbf{w}\|_2 \leq \lambda_k$ and the number of internal nodes bounded by $n_k$. The regularized objective $L(\boldsymbol{\alpha})$ as in Equation 7 has Lipschitz-continuous derivatives with the coordinate-wise Lipschitz constants $C_k$ bounded as follows:*

$$C_k \leq \lambda_k \left[ \max_{1 \leq i \leq m} C_\Phi(y_i) \right]. \tag{25}$$

*Proof.* For a sample $S$ and a fixed tree $h$ let $\eta_l$ be the number of sample points falling within the leaf $l$.

$$C_k \leq \frac{1}{m} \left[ \max_{1 \leq i \leq m} C_\Phi(y_i) \right] \sum_{i=1}^{m} h_k^2(x_i)$$

$$\leq \frac{1}{m} \left[ \max_{1 \leq i \leq m} C_\Phi(y_i) \right] \sum_{l \in \text{leaves}(h_k)} \eta_l w_l^2$$

$$\leq \frac{1}{m} \left[ \max_{1 \leq i \leq m} C_\Phi(y_i) \right] \|\mathbf{w}\|_2 \max_{l \in \text{leaves}(h_k)} \eta_l$$

$$\leq \|\mathbf{w}\|_2 \left[ \max_{1 \leq i \leq m} C_\Phi(y_i) \right]$$

$$\leq \lambda_k \left[ \max_{1 \leq i \leq m} C_\Phi(y_i) \right]$$

This results in the coordinate sampling distribution for the Randomized Coordinate Descent.

$$p_k = \frac{\lambda_k}{\sum_{j=1}^{K} \lambda_j} \tag{26}$$

$\square$

**Table 2:** Dataset statistics

|          | sonar | cancer | diabetes | ocr17 | ocr49 | mnist17 | mnist49 |
|----------|-------|--------|----------|-------|-------|---------|---------|
| Examples | 208   | 699    | 768      | 2000  | 2000  | 15170   | 13782   |
| Features | 60    | 9      | 8        | 196   | 196   | 400     | 400     |

## A.5 Descriptive statistics of the UCI datasets

Note that mnist17 and mnist49 refer to the original 20-by-20 pixel datasets, where only two digits (1,7 and 4,9 respectively) were sampled. The *cancer* dataset refers to the *breastcancer* dataset in the UCI repository.