[Reviews · NeurIPS 2019]

Reviewer 1



The paper is well organized and written. First an important problem is discussed, then theoretical bounds are given to address this problem, driven by which a new algorithm is proposed. However, there are some questions need more explanations. (1) Why do you restrict the analysis to the hypothesis families of regression trees and say your results can be extended to other families? (2) Although it says running time of RGB per thread is comparable to that of standard GB, I’m worried about the efficiency of RGB. It would be better if running time of the experiments are given and compared. (3) There is only one algorithm GB compared with RGB. If RGB is able to beat many other algorithms across different datasets, it would be considered to be more practical.

Reviewer 2



originality: Incremental. Guided by the theoretical results, this paper proposes to find a function by Randomized Coordinate Descent in each step of boosting. quality: The theoretical part and the proposed algorithm are sound. However, the proposed RGB is much slower than the original GB. clarity: This paper is well written. However, the theoretical part is not easy to follow. significance: Not clear, due to the experiments are in small toy data (UCI data), no results on large datasets. cons: 1. My most concern is the speed of RGB for it will search multiple trees in each iteration. And I do not think Line 271-273 is correct, for the single tree learning in GBDT could be in parallel, such as XGBoost and LightGBM. Therefore, RGB will be much slower than GB. And experiment should have comparison over speed as well. 2. The used datasets in this paper are all small datasets from UCI. It would be better to have a comparison on large datasets, for we don't use small data in most real-world tasks.

Reviewer 3



Gradient boosting (GB) has been extensively studied in the past, both theoretically and experimentally. Recently, with the advent of big data, several accelerated versions of vanilla GB have been proposed (in particular the well known XGBoost), and while the experimental evaluations of these methods have been abundant, the same cannot be said for the theoretical analysis. In this paper, the authors tackle this important problem. The main contribution of this paper consists in casting the various accelerated GB methods in a regularized gradient boosting setting. Indeed, by introducing a regularization term in the usual minimization objective of GB, it is possible to recover most, if not all, of the various accelerated gradient boosting approaches (XGboost included), while at the same time opening up several interesting and exciting possibilities for deriving new/novel acceleration strategies. The second contribution is a full theoretical analysis of the generalization error of regularized gradient boosting using the Rademacher complexity of the hypothesis space. To the best of my knowledge, this is the first bound on the generalization error custom tailored for accelerated gradient boosting approaches. The third contribution comes in the form of a theoretically justified learning algorithm, coined RGB. The authors make perfect use of the theoretical guarantees of randomized coordinate descent in order to justify and derive the novel method RGB. The paper is also very well written and easy to follow. The main contributions are clearly presented. As far as I could tell, the theoretical justifications are correct. My main remarks concern the experimental section. While I agree that the experimental results are clearly encouraging, as they show a certain advantage of RGB over XGBoost, I'm not quite sure I clearly understand the authors' claim in lines 307-311. In particular, I don't think that such bold conclusions can be inferred from the results presented in the paper, unless I'm missing something. Also, in line 303 the authors claim that only RGB fails only on diabetes, yet the results of Table 1 show that RGB fails on mnist49 as well, which implies that the hypothesis given in lines 304-306 is not realistic. All in all, a very interesting contribution, proposing a theoretical view of accelerated (regularized) gradient boosting.

[Author Response · NeurIPS 2019]

We thank all the reviewers for their comments. We first address questions raised by multiple reviewers and next respond to individual questions.

Running time and efficiency of RGB. To address the RGB runtime questions raised by multiple reviewers, we provide a very simple explanation. Given: a) a fixed sample of $S$ coordinates (Algoritm 1, line 2.) b) the same subroutine is used for single tree node splitting and pruning; then the runtime of one RGB round is equal to the runtime of S rounds of GB. If $S = 1$, then the runtime of RGB is equal to that of GB. Given $S$ parallel workers, the time per worker of RGB is equal to that of GB. The measured runtime of our experiments confirms the explanation above and we are happy to include the runtime plots in the paper. Since our algorithm within a single boosting round allows to sample and evaluate each candidate in parallel, the training is efficient given multiple workers.

Experimental benchmarking of RGB against other algorithms. As some reviewers suggested it is definitely possible to compare RGB with ensemble algorithms other than GB (e.g. reinforcement learning, random forest, etc). We consciously made a choice to provide comparison with GB only, and specifically for the families of regression trees, since it most clearly illustrates our novel methodology. This is because both these algorithms explore similar hypotheses spaces $\mathcal{H}$ and we would like to show that RGB by using generalization bound and randomization makes use of $\mathcal{H}$ better than GB. It would not be a fair or a meaningful experiment to compare RGB with ensembles of neural networks for example, since the underlying spaces $\mathcal{H}$ are not comparable. It is important to stress that we don't claim that RGB beats all other ensemble algorithms which can vary in hypotheses class complexity and search methods, we rather claim that given the same hypotheses space $\mathcal{H}$, RGB does better than a standard gradient boosting algorithm by making the use of generalization properties of $\mathcal{H}$ and random search over $\mathcal{H}$.

Experimental benchmarking of RGB for other datasets. We are happy to include a variety of experiments on larger datasets in the final version. We already confirmed that the results on UCI and MNIST datasets carry over to Higgs dataset. Overall, since this submission presents novel theory, methodology and algorithm, this is not a purely empirical paper and the UCI as well as MNIST dataset experiments that we presented serve as an illustration of the power of our approach.

REV1. "Why do you restrict the analysis to the hypothesis families of regression trees...?". First, regression trees is the most widely used hypothesis family in the boosting literature; second, doing similar generalization analysis for other families such as neural networks would involve the same steps, but a different and nontrivial proofs for the Rademacher complexity bounds, which is beyond the scope of this work.

REV2. "My most concern is the speed of RGB for it will search multiple trees in each iteration. And I do not think Line 271-273 is correct". Indeed, one tree fitting for GB can be done in a multi-threaded way, but not necessarily since for example in XGBoost does not support exact tree splitting method distributed. However, for our experiments whenever we use a multi-threaded subroutine for single tree splitting, we use it both in RGB and GB, making the runtime comparison consistent. Thus, it is correct that one round of RGB takes as much time as $S$ rounds of GB, which is also explained above.

"(Significance) Not clear, due to the experiments are in small toy data (UCI data), no results on large datasets." We addressed this above. We would like to add that this work has meaningful contributions in theory and methodology, not just experiments - as nicely summarized by REV3. For example, we provide the missing theory for regularized boosting, that explains the impact of regularization on its generalization properties. Thus, the purpose of the experiments that we provide is to illustrate the power of the novel methodology, but not to win the state of the art across many tasks.

"More experiments on large datasets, not UCI...". We would like to bring to the Reviewer's attention that we have presented experiments on larger MNIST datasets in addition to the UCI datasets. As suggested, we are happy to include a variety of large scale experiments in the final version and we have confirmed that the MNIST results carry over to the Higgs dataset.

REV3. "...the authors' claim in lines 307-311...". We would like to explain to the Reviewer, that the claim we make is more specific. We claim that compared to an algorithm that operates on a hypotheses space $\mathcal{H}$ of the same complexity, RGB will reduce over-fitting by leveraging the theory that we developed. This is precisely captured by our experiment presented. However, if we compare different algorithms on widely varying hypotheses spaces, we would not be able to draw a conclusion. For example, we can't make a claim about RGB on regression trees with depth up to 10 versus ensembles of neural networks with 10 layers.

[Meta-Review · NeurIPS 2019]

This paper proposes Rademacher generalization bounds for Regularized Gradient Boosting which encompasses various accelerated GB methods. Although there are still some work to be done in order to make the proposed algorithm derived from the theoretical study faster but the proposed theoretical study deserves publication.